# Determinants of neonatal near miss among neonates admitted to public hospitals in Southern Ethiopia, 2021: A case-control study

**Aklilu Habte**[1]*, **Kaleegziabher Lukas**[1], **Tamirat Melis**[2], **Aiggan Tamene**[1], **Tadesse Sahle**[3], **Mulugeta Hailu**[1], **Addisalem Gizachew**[1]

**1** School of public health, College of Medicine and Health Sciences, Wachemo University, Hosanna, Ethiopia, **2** Department of public health, College of Medicine and Health Sciences, Wolkite University, Wolkite, Ethiopia, **3** Department of Nursing, College of Medicine and Health Sciences, Wolkite University, Wolkite, Ethiopia

* akliluhabte57@gmail.com

**Data Availability Statement:** All relevant data are within the paper and its Supporting Information files.

## Abstract

### Background

Neonatal near-miss (NNM) cases refer to situations in which babies are on the verge of dying between the ages of 0 and 28 days due to severe morbidity that occurs during pregnancy, delivery, or extra-uterine life, but survive either by luck or due to high-quality health care. Identifying NNM cases and addressing their determinants is crucial for devising comprehensive and relevant interventions to tackle neonatal morbidity and mortality. Hence, this study aimed at finding out the determinants of NNM in neonates admitted to public hospitals in Hadiya zone, southern Ethiopia.

### Methods

A hospital-based unmatched case-control study was conducted in three selected hospitals in southern Ethiopia from May 1 to June 30, 2021. A total of 484 participants took part in the study (121 cases and 363 controls). Controls were chosen using systematic sampling approaches, whereas cases were recruited consecutively at the time of discharge. Cases were selected based on the Latin American Centre for Perinatology (CLAP) criteria of an NNM. A structured interviewer-administered questionnaire and a data extraction checklist were used for data collection. The Data were entered into Epi-Data version 3.1 and exported to SPSS version 23 for analysis. A multivariable logistic regression analysis with a p-value of <0.05 was used to determine the determinants of NNM.

### Results

Ninety-seven (80.1%) and 56 (46.2%) near-miss cases encountered at least one pragmatic and management criteria, respectively. The most common pragmatic and management criteria were gestational age less than 33 weeks (44.6%) and intravenous antibiotic usage up to 7 days and before 28 days of life (27.3%), respectively. A short birth interval [AOR = 2.15, 95% CI: 1.29, 3.57], lack of ANC [AOR = 3.37; 95%CI: 1.35, 6.39], Caesarean mode of

**Funding:** The author(s) received no specific funding for this work.

**Competing interests:** The authors have declared that no competing interests exist.

**Abbreviations:** ANC, antenatal care; AOR, adjusted odds ratio; BPCR, birth preparedness and complication readiness; CLAP, Latin American Center of Perinatology; CS, Cesarean section; EDHS, Ethiopia demographic and health survey; NMR, Neonatal mortality ratio; NNM, neonatal near miss; WHO, World Health Organization.

delivery [AOR = 2.24; 95%CI: 1.20, 4.16], the occurrence of a third maternal delay [AOR = 3.47; 95% CI: 2.11, 5.75], and poor birth preparedness and complication readiness (BPCR) plan[AOR = 2.50; 95% CI: 1.49,4.13] were identified as a significant determinants of NNM.

## Conclusion and recommendation

The provision of adequate ANC should be a priority for health care providers at service delivery points. To avoid serious neonatal problems, mothers who deliver by Cesarean section should receive more attention from their families and health care providers. Health care providers in the ANC unit should encourage pregnant women to implement the WHO-recommended elements of the BPCR plan. To achieve optimal birth spacing, healthcare providers should focus on the contraceptive provision. Unnecessary delays in health facilities during childbirth should be avoided at all costs.

## Background

Neonatal mortality has long been regarded as a key indicator of social, economic, and healthcare advancements [1]. About one-third and three-quarters of neonatal deaths in the first month of life occur on the day of birth and the first week of life, respectively [2,3]. Between 1990 and 2017, global statistics revealed a 51% decrease in death; nevertheless, the fall in early neonatal mortality rate(NMR) has been slower than the decline in post-neonatal under-five mortality [4,5]. According to a global estimate in 2017, more than 2.7 million children under the age of five died, with almost one million (37%) of these deaths occurring in neonates within the first seven days of life outside the womb [6].

Neonatal death (NND) is most common in developing countries(98%), with the majority of cases occurring at home and outside of the formal healthcare system [7]. This figure was dominated by countries in South Central Asia and Sub-Saharan Africa [8,9]. A child born in Sub-Saharan Africa (SSA) is ten times more likely than a child born in a high-income nation to die in the first month [9,10]. Just five countries, Ethiopia, Nigeria, the Democratic Republic of Congo, Tanzania, and Uganda, have experienced half (50%) of neonatal mortality in this region [9]. Although NMR in both developed and developing countries are declining, severe neonatal morbidities like neonatal near-miss cases remain high [11].

Neonatal near-miss(NNM) refers to situations in which neonates are on the verge of dying between the ages of 0 and 28 days due to severe morbidity that occurs during pregnancy, delivery, or extra-uterine life, but survive either by luck or due to high-quality of care [12,13]. After reviewing various studies on NNM, the Latin American Centre for Perinatology (CLAP) and the Pan American Health Organization developed a standardized definition that defined NNM as any newborn infant who encountered at least one of the pragmatic and/or management criteria and survived the first 27 days of life [12,14,15].

The pragmatic criteria are a birth weight of <1750 grams, an APGAR score of less than 7 at 5 minutes of life, and gestational age of <33 weeks. Parenteral therapeutic antibiotics, nasal continuous positive airway pressure, any intubation during the first 27 days of life, phototherapy within the first 24 hours of life, cardiopulmonary resuscitation, vasoactive drugs, anticonvulsants, surfactants, blood products, and steroids for refractory hypoglycemia, and any surgical procedure are among the management criteria used. They also recommended some management criteria that have not been studied before, such as the use of an antenatal steroid, parenteral feeding, congenital deformity, and admission to the NICU [12,14,15].

Most neonatal death that occurred worldwide were due to the pragmatic criteria component of NNM cases [16]. Globally, birth asphyxia and preterm complications accounted for 24% and 35% of neonatal deaths, respectively [17]. Similarly, 14% of newborns delivered worldwide were underweight, with Asian and African countries having the greatest rates [18,19]. These conditions have long-term consequences on neurological and cognitive development. They have also been attributed to cardiovascular disease, pulmonary disease, and severe disabilities such as blindness or low vision, as well as hearing loss [20,21]. These resulted in a significant psychological, emotional, and financial strain on the family, society, and the patient [22].

Because of the different criteria utilized within every study, the level of neonatal near miss (NNM) differed greatly. According to certain studies, the number of neonates who survived severe morbidities was roughly 3 to 6 times higher than those who died [15,23,24]. According to studies based solely on pragmatic criteria, the incidence of NNM ranged from 21.4/1000 live births in Brazil [25] to 86.7/1000 live births in India [26]. Whereas, according to those studies done by combining both pragmatic and management criteria, the figure ranged from 39.2/1000 live births [27] to 367/1000 live births [28]. Maternal education, parity, antepartum hemorrhage, hypertensive disorders of pregnancy (HDP), history of low birth weight, and frequency of ANC visits have all been identified as determinants of NNM in studies conducted around the world, including Ethiopia [14,24,29–34].

Currently, the Global Maternal and Child Survival Program focus on newborns in developing countries, like Ethiopia, by implementing a community-based newborn care effort that improves maternal and neonatal healthcare-seeking behavior by identifying and treating sepsis [35,36]. Despite all of FMOH's efforts, Ethiopia's infant mortality rate has risen from 29/1000 LB to 30/1000 LB [37,38].

The near-miss concept and criterion-based clinical audit are two novel ideas for gathering critical information in neonatal care and improving prenatal care quality [13]. Assessing cases of neonatal near-misses and addressing contributing factors can provide a comprehensive and relevant approach to preventing neonatal death [23,28,39]. There was limited research on the determinants of NNM in Ethiopia, and none in the current study area. Although some studies were done in Ethiopia [33,34], they duly emphasized maternal factors and failed in identifying health system-related factors like the impact of the three-maternal delays and Birth preparedness, and complication readiness plan(BPCRP). Hence, this study aimed at identifying the proximate, intermediate, and distal determinants of NNM among neonates admitted to public hospitals in Hadiya Zone, southern Ethiopia. The findings might help to program managers and planners in identifying the factors that contribute to NNM, allowing them to take appropriate interventions to tackle neonatal morbidity and mortality.

## Methods and materials

### Study area, period, and design

From May 1 to June 30, 2021, a facility-based unmatched case-control study was conducted at selected public hospitals in the Hadiya zone, Southern Ethiopia. The Zone is one of the 17 zones in the Southern Nations, Nationalities, and Peoples' Region (SNNPR) of Ethiopia with 13districts, 4 town administrations. Hossana town, the zone's capital, is 230 kilometers from, Addis Ababa, the capital city of Ethiopia. According to the report of the Zonal Health Department, the total population in 2020 was 1,797,395 (Male = 893,594, Female = 903,801). Regarding health facilities, there was one general hospital, three primary hospitals, 59 government health centers, and 311 health posts. The estimated number of reproductive-age women (15–49) and live births were 470,587 and 64,608, respectively.

## The population of the study

All neonates admitted to public hospitals in the Hadiya zone constituted the source population whereas the study populations were selected neonates admitted to selected public hospitals in the Hadiya zone during the study period. Cases were selected by applying the Latin American Centre for Perinatology (CLAP) definition for a neonatal near miss. NNM events were considered when the newborn faced at least one of the near-miss criteria or exhibited pragmatic and/or management criteria but survived this condition within the first 27 days of life. Pragmatic criteria are Birth weight < 1750g, gestational age < 33 weeks, 5th-minute Apgar score < 7 whereas management criteria are: parenteral therapeutic antibiotics; nasal continuous positive airway pressure(NCPAP); any intubation during the neonatal period, phototherapy within the first 24 hours of life, cardiopulmonary resuscitation, the use of vasoactive drugs, anticonvulsants, surfactants, blood products, and steroids for refractory hypoglycemia and any surgical procedure [12,13,32,40]. Healthy neonates (without complications) who were admitted to the post-natal or neonatal ward by a pediatrician, neonatologist, gynecologist, or resident as a healthy babies were used as controls. Three controls were selected for each near-miss case on the same day as the near-miss event.

Those neonates who were delivered at home, referred from other health care facilities (outside of selected hospitals), were multiple births (twins), and were initially selected as a control and discharged but returned as a case during the study period were excluded. Furthermore, neonates who were not with their mothers or whose mothers' histories were unknown during the study period were also excluded.

## Sample size determination

The sample size for the study was determined by applying a double population proportion formula through Epi Info 7 stat calc program. The following assumptions were put into consideration: Confidence level of 95%, power of the study of 80%, the case-control ratio of 1:3, percent of exposure among case and control. To get the maximum sample size, various covariates across multiple studies conducted in different countries were considered (Table 1). The percentage of cases exposed to old maternal age (5.4%) and the percentage of controls exposed to old maternal age (15.8%) were taken from a study conducted in Brazil [12]. Based on the above assumptions the estimated sample size was 440 (110cases and 330controls). After considering the nonresponse of 10%, the final sample size used for this study was 484(121 cases and 363 controls).

## Sampling procedures

From a total of four hospitals in the Hadiya zone, three (Wachemo university Nigist Eleni Mohammed Memorial comprehensive and specialized hospital, Shone General Hospital, and Gimbichu Primary hospital) were selected randomly. The total number of cases and controls admitted in each hospital during the previous fiscal year in two consecutive months (May and

**Table 1. Sample size determination for determinants of neonatal near miss (NNM) among neonates admitted to public hospitals in Southern Ethiopia, 2021.**

| Sr. No | Variables | CI | Power | % of exposure among cases | % of exposure among controls | sample size after adding None response rate of 10% | Ref. |
|---|---|---|---|---|---|---|---|
| 1 | Having <4ANC visit | 95% | 80% | 6.2 | 29.2 | 40+118 = 158 | [12] |
| 2 | Maternal Age is greater than 35 yrs. | 95% | 80% | 5.4 | 15.8 | 121+363 = 484 | [12] |
|  |  | 95% | 80% | 27.6 | 13.4 | 97+291 = 388 | [33] |
| 3 | Instrumental delivery | 95% | 80% | 26.9 | 6.7 | 49+146 = 195 | [33] |

June) were counted from registrations, and the average had been used as a baseline. Afterward, the proportional allocation has been used to determine the sample size for each hospital. Finally, cases were selected consecutively at discharge until the required sample size was attained (Fig 1).

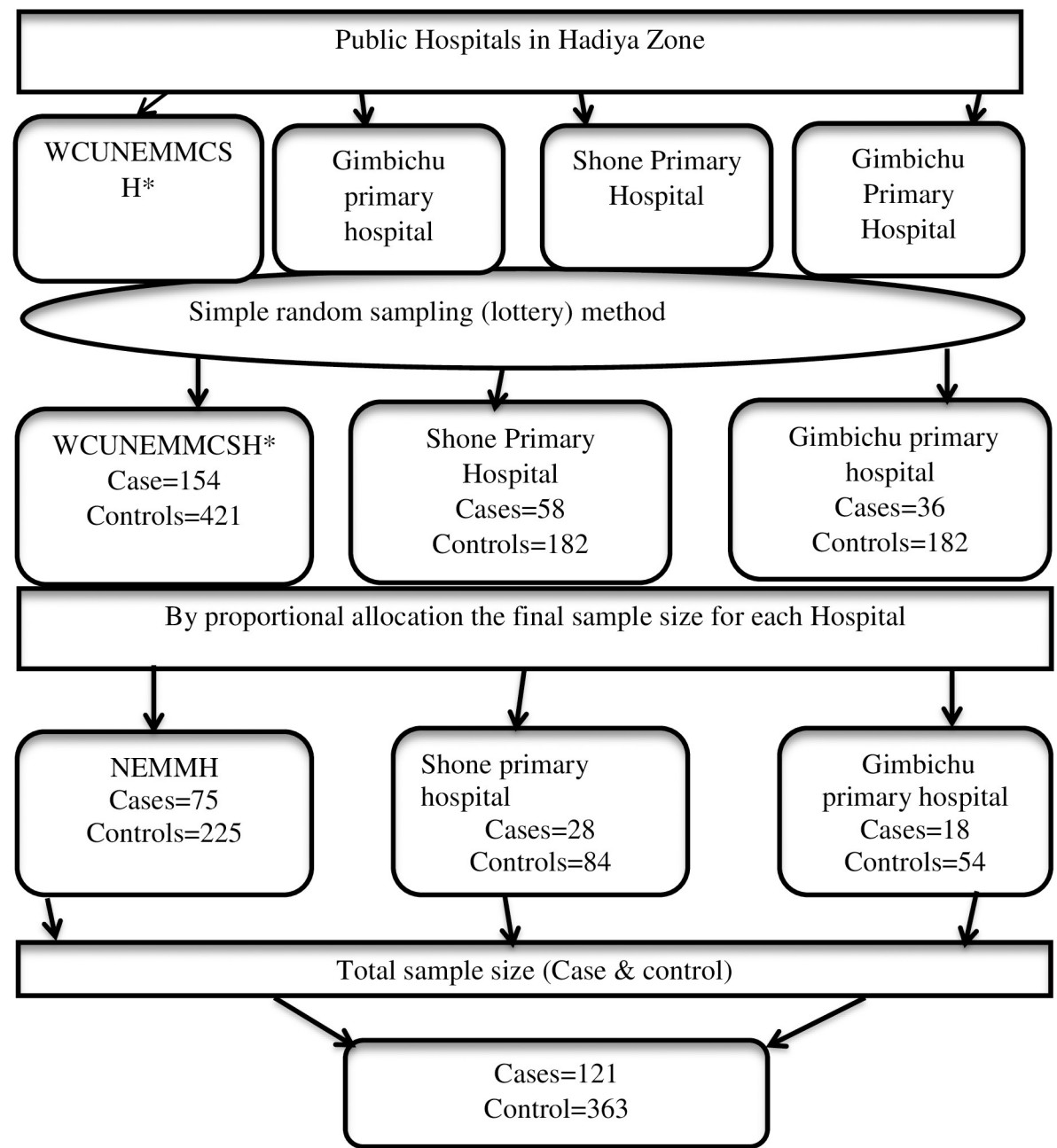

*Wachemo university Nigist Eleni Mohammed Memorial Comprehensive and Specialized Hospital

**Fig 1. Schematic presentation of sampling procedures followed to get study participants in public hospitals of Hadiya zone, southern Ethiopia, 2021.**

## Data collection tools, methods, and personnel

A two way of data collection was used. The data from the mother's side were collected using a pretested, structured, and interviewer-administered questionnaire adapted from relevant pieces of literature [12,14,15,33]. The questionnaire was specifically designed to collect data on socio-demographic factors, obstetric factors, and medical conditions during pregnancy, as well as newborn-related characteristics and healthcare system-related characteristics. The socioeconomic status of households was determined using a tool adapted from the 2016 EDHS, which consisted of 36 items grouped as follows: household assets, livestock ownership, crop production in quintals, average estimated monthly income, agricultural land ownership in hectares, and residential home with its infrastructures [38]. Six well-trained BSc midwives with data collection experience and fluency in the local languages collected data under the supervision of three BSc holder nurses. A data abstraction checklist was used to collect information on NNM events from the medical records of neonates.

## Data quality management

After translation into the local language, Amharic, properly designed data collection tools were provided. The principal investigator provided the data collectors and supervisors a two-day intensive training on the technique of timely data collection, the purpose of data collection, the contents of the questionnaires, how to approach the respondents, and the issue of confidentiality and privacy. One week before the actual data collection, a pretest was conducted on 5% of the sample size (6 cases 19 controls) at Worabe comprehensive and specialized hospital to assess the validity of the data collection tool. All the necessary corrections were made based on the pretest result. The reliability of the questionnaires was assessed and found to be good, with a reliability index (Cronbach's alpha) of 0.81. Those health care providers in MNCH departments (delivery ward, postnatal ward, and NICU) of each hospital were informed of the study and told to notify the data collectors if they get near-miss case/s. In addition, the criteria for NNM case identification were posted on the wall of each ward. During the data collection period, the principal investigator and supervisors conducted on-site supervision. Every day, the supervisors and principal investigator read and checked each questionnaire for completeness, and the necessary comments were given to the data collectors before the next day. To reduce social desirability bias, study participants were interviewed in private.

## Definition and operationalization of variables

Cases: Were those neonates got survived despite being exposed to at least one of the proposed criteria. From pragmatic criteria: Birth weight < 1750g, gestational age < 33 weeks, 5th-minute Apgar score < 7 and/or from the management criteria: parenteral therapeutic antibiotics; nasal continuous positive airway pressure; any intubation during the first 27 days of life; phototherapy within the first 24 hours of life; cardiopulmonary resuscitation; the use of vasoactive drugs, anticonvulsants, surfactants, blood products and steroids for refractory hypoglycemia and any surgical procedure [12,14,33]. Cases and controls were ascertained based on the initial diagnosis made by higher experts like pediatricians, neonatologists, gynecologists, or residents in maternal and child health specialties.

Controls (a healthy neonate): is defined as any baby born with the best extra uterine life adaptation (APGAR > 7) and no clinically apparent malformation.

APGAR score: is a score ranging from 0–10 based on a newborn's tone, color, respiration, pulse rate, and responsiveness at 1, 5, and 10 minutes and 7–10 scores of this variable indicate that a healthy baby and 0–6 indicate distressed neonates.

Birth weight: was defined as Very low birth weight <1500gm, low birth weight 1500-2500gm, normal birth weight 2500-4000gm, and macrosomia ≥4000gm.

Gestational age: Gestational age has been defined as Preterm if GA<37, Term if GA = 37–42, and Post-term if GA>42 weeks [41].

Maternal complication: Those mothers come with one of the following compliance: Obstructed labor, hypertensive disorders of pregnancy, Hemorrhage, Sepsis, and Others [41].

Being model household (MHH): Those participants who were implementing all health extension packages and got a certificate of appreciation from concerned bodies [42].

A good Birth Preparedness and Complication Readiness(BPCR) plan: Described as a woman who implemented at least five of the eight WHO recommendations: ascertained birthplace and birth attendants, established emergency transportation; put the money asides, identified labor, and birth companion; identified nearest health institution; identified blood donors if necessary, and identified care provider for children at home while the mother was away [43,44].

Knowledge of key newborn danger signs: The nine WHO-UNICEF lists of newborn danger signs have been used to assess mothers' knowledge of these signs, which included inability to feed since birth or stop feeding, convulsions, fast breathing, severe chest in-drawing, high-grade fever, cold extremities, only moves when stimulated, or not even when stimulated, yellowish discoloration of extremities, and signs of local infection (umbilicus red or draining pus, skin boils, or eyes draining pus) [45]. A woman who scored above the mean was deemed knowledgeable; if she did not, she was considered as not knowledgeable.

The first maternal delay: was the period between the identification of health problems and decision-making to pursue maternal health care. A delay was deemed when it take more than 24 hours to decide to seek treatment, otherwise was no delay [43].

Second maternal delay: was a time after decision-making to reach health facilities. The time has been estimated at more than one hour to reach the existing health facility and otherwise not [43].

Third maternal delay: is a delay in receiving care in health facilities and is measured by the time interval between reaching the health facility and accessing the required services. When it took more than 1 hour it was deemed as a delay otherwise no delay [43].

Autonomy to maternity care. This is how resources are identified and controlled when women should seek maternal health services and classified as: autonomous, if she decides alone or with her husband (jointly) to seek maternal and child health care; otherwise not autonomous, it means a husband alone or a third party decided on the use of the services [42].

## Data analysis

The data were entered into Epi-Data version 3.1 and exported to SPSS version 23 for analysis. Running frequencies were used to check for inconsistencies and missing data. Univariate analyses including frequency, proportion, mean, and standard deviation were calculated for both cases and controls. The principal component analysis (PCA) was conducted to examine the wealth index of each household. Initially, 36 items were used to measure the wealth status of participants, including household assets, livestock ownership, crop production in quintals, average estimated monthly income, agricultural land in hectares, and residential house with their infrastructures. If the asset or variables were owned by more than 95% of the sample or less than 5% of the sample, they were removed. Kaiser-Meyer-Olkin measure of sampling adequacy (≥ 0.6), Bartlett's Test of Sphericity (p-value < 0.05), and anti-image correlations (> 0.4) for sampling adequacy of individual variables were checked for the fulfillment of assumptions for PCA. Those variables with communalities less than 0.5 and complex

structures (i.e. having correlations > 0.4 in more than one component) were removed iteratively until the assumptions were fulfilled.

The Chi-square test was used to compare the proportion of cases and controls between selected categorical variables. Bivariable and multivariable logistic regression analyses were done to identify the determinants of NNM. In the bivariable analysis, explanatory variables with p-values less than 0.25 were eligible for multivariable logistic regression. Finally, determinants of NNM were determined in the final model with a p-value of <0.05 and a 95%CI with AOR. Model fitness was measured using the Hosmer and Lemeshow goodness of fit tests, and the Nagelkerke R Square, which were 0.64 and 0.548, respectively. The variance inflation factor (VIF) was used to check for multicollinearity amongst the explanatory variables and was 7 which is <10.

### Ethical consideration and consent to participate

The Institutional Review Board (IRB) of Wachemo University College of Medicine and Health Science granted written Ethical clearance. The study's purpose and procedures were explained to the participants. Participants aged 18 and up signed a written informed consent form. Furthermore, for those participants under the age of 18, consent was obtained from a parent or guardian using standard disclosure procedures. A unique ID number was issued to the questionnaire to maintain its confidentiality. Participants' privacy and confidentiality were guaranteed before data collection.

## Results

### Socio-demographic characteristics of respondents

A total of 121 cases and 363 controls took part in the study yielded a response rate of 100% for both. The mean (±SD) age for neonates' mothers was 29.9 (±4.6) years for cases and 30.0 (±5.0) years for controls. However, the mean age difference between cases and controls was not statistically significant when examined by using the Chi-square test. Rural residents made up 67 (55.4%) of the case group and 131 (36.1%) of the controls group respondents. In terms of educational status, 46 (38.0%) and 116 (31.9%) of respondents in the case and control groups, respectively, did not receive a formal education. In comparison to controls, a large proportion (22.3%) of cases were from families in the lowest quintile of wealth (17.1%) (Table 2).

### Characteristics of the newborns

With 62(51.2%) and 59(48.8%), respectively, male and female cases were almost equally represented. The Chi-square tests showed that there were no statistically significant differences in sex across cases and controls. A vertex presentation was seen in the majority of cases 91 (75.2%) and controls 314(86.5%). Furthermore, the proportion of non-vertex presentation was higher among cases 30(24.8%) than in controls 43(11.8%).

### Obstetric characteristics of respondents

In the cases and controls groups, respectively, 73 (60.3%) and 207 (57.0%) of the respondents were multiparous (birth order 2–4). History of stillbirth was reported by mothers of 7(5.8%) cases and 34(9.4%) of controls. Seventeen (14.0%) and 46 (12.7%) of mothers in the cases and control groups, respectively, had had a history of abortion. Among women who gave birth within <24-month interval, the proportions of cases and controls were 67 (55.4%) and 114 (31.4%), respectively. Eighteen (14.9%) and 26 (7.2%) of mothers of cases and controls, respectively, had a history of neonatal death (Table 3).

**Table 2. Socio-demographic characteristics of mothers of neonates admitted to Public Hospitals in Hadiya Zone, Southern Ethiopia, 2021.**

| Variable categories | Cases = 121 | Controls = 363 | Total = 484 | Test statistics |
|---|---|---|---|---|
| | [n(%)] | [n(%)] | [n(%)] | |
| **Age of mother in years** | | | | |
| 35+ | 19(15.8) | 59(16.3) | 78(16.1) | χ2 = 1.692 |
| 20–34 | 97(80.1) | 277(76.3) | 374(77.3) | p = 0.429 |
| <20 | 5(4.1) | 27(7.4) | 32(6.6) | |
| **Residence** | | | | |
| Urban | 54(44.6) | 232(63.9) | 286(59.1) | χ2 = 13.960 |
| Rural | 67(55.4) | 131(36.1) | 198(40.9) | P <0.001 |
| **Marital status** | | | | |
| In marital union | 111(91.7) | 338(93.1) | 449(92.8) | χ2 = 0.257 |
| Not in marital relation | 10(8.3) | 25(6.9) | 35(7.2) | P = 0.612 |
| **Religion** | | | | |
| Orthodox | 31(25.6) | 120(33.0) | 151(31.2) | χ2 = 4.708 |
| Protestant | 52(43.0) | 140(38.6) | 192(39.7) | P = 0.127 |
| Muslim | 30(24.8) | 93(25.6) | 123(25.4) | |
| Catholic | 8(6.6) | 10(2.8) | 18(3.7) | |
| **Ethnicity** | | | | |
| Hadiya | 91(75.2) | 290(79.9) | 381(78.7) | χ2 = 4.708 |
| Kembata | 22(18.2) | 58(16.0) | 80(16.5) | P = 0.127 |
| Siltie | 5(4.1) | 8(2.2) | 13(2.7) | |
| Others | 3(2.5) | 7(1.9) | 10(2.1) | |
| **Mother's Educational level** | | | | |
| No formal education | 46(38.0) | 116(31.9) | 162(33.5) | χ2 = 7.373 |
| Primary education (1-8th) | 31(25.6) | 87(24.0) | 118(24.4) | P = 0.061 |
| Secondary(9-12th) | 30(24.8) | 87(24.0) | 117(24.1) | |
| College and above | 14(11.6) | 73(20.1) | 87(18.0) | |
| **Husband's Education(n = 449)** | | | | |
| No formal education | 26(23.2) | 75(22.2) | 101(22.5) | χ2 = 0.912 |
| Primary education (1-8th) | 39(34.8) | 105(31.1) | 144(32.1) | P = 0.823 |
| Secondary(9-12th) | 22(19.6) | 73(21.6) | 95(21.1) | |
| College and above | 24(21.4) | 85(25.1) | 109(24.3) | |
| **Wealth index** | | | | |
| Highest | 19(15.7) | 79(21.8) | 98(20.2) | χ2 = 5.085 |
| Fourth | 19(15.7) | 76(20.9) | 95(19.6) | P = 0.279 |
| Middle | 27(22.3) | 72(19.8) | 99(20.5) | |
| Second | 29(24.0) | 74(20.4) | 103(21.3) | |
| Lowest | 27(22.3) | 62(17.1) | 89(18.4) | |
| Family size | | | | |
| <5 | 56(46.3) | 183(50.4) | 239(49.4) | χ2 = 0.620 |
| ≥5 | 65(53.7) | 180(49.6) | 245(50.6) | P = 0.431 |

## Maternal health service-related characteristics

Sixteen (13.2%) and 18 (5.0%) of mothers in the cases and control groups, respectively, had no antenatal care (ANC) follow-up. The control group had a higher percentage of mothers (41.3%) who had four or more ANC visits than the cases group (24.8%). In terms of mode of delivery, 82(16.9%) of neonates' mothers gave birth by cesarean section, with 36 (29.7%) from

**Table 3. Obstetric characteristics of mothers of neonates admitted to Public Hospitals in Hadiya Zone, Southern Ethiopia, 2021.**

| Variable categories | Cases = 121 | Controls = 363 | Total = 484 | Test statistics |
|---|---|---|---|---|
| | n(%) | n(%) | n (%) | |
| **Gravidity** | | | | |
| 1 | 12(9.9) | 29(8.0) | 41(8.5) | $\chi2 = 0.505$ |
| 2–4 | 74(61.1) | 222(61.1) | 296(61.1) | P = 0.775 |
| ≥5 | 35(30.0) | 112(30.9) | 147(30.4) | |
| **Parity** | | | | |
| 1(Primipara) | 17(14.1) | 66(18.2) | 83(17.1) | $\chi2 = 1.100$ |
| 2-4(Multipara) | 73(60.3) | 207(57.0) | 280(57.9) | P = 0.577 |
| ≥5(Grand multipara) | 31(25.6) | 90(24.8) | 121(25.0) | |
| **Birth interval** | | | | |
| ≥24 months | 54(44.6) | 249(68.6) | 303(62.6) | $\chi2 = 22.266$ |
| <24 months | 67(55.4) | 114(31.4) | 181(37.4) | P<0.001 |
| **Desire on the last pregnancy** | | | | |
| Unplanned | 37(30.6) | 84(23.1) | 121(25.0) | $\chi2 = 2.678$ |
| Planned | 84(69.4) | 279(76.9) | 363(75.0) | P = 0.102 |
| **History of stillbirth** | | | | |
| Yes | 7(5.8) | 34(9.4) | 41(8.5) | $\chi2 = 1.501$ |
| No | 114(94.2) | 329(90.6) | 443(91.5) | P = 0.220 |
| **History of Neonatal death** | | | | |
| No | 103(85.1) | 337(92.8) | 440(90.9) | $\chi2 = 6.533$ |
| Yes | 18(14.9) | 26(7.2) | 44(9.1) | P = 0.220 |
| **Ever had abortion** | | | | |
| Yes | 17(14.0) | 46(12.7) | 63(13.0) | $\chi2 = 0.152$ |
| No | 104(86.0) | 317(87.3) | 421(87.0) | P = 0.697 |
| **Frequency of abortion (n = 63)** | | | | |
| Once | 5(29.4) | 19(41.3) | 24(38.1) | $\chi2 = 0.272$ |
| More than once | 12(70.6) | 27(58.7) | 39(61.9) | P = 0.797 |
| **Ever had a history of preterm birth** | | | | |
| Yes | 6(5.0) | 27(7.4) | 33(6.8) | $\chi2 = 0.878$ |
| No | 115(95) | 336(92.6) | 451(93.2) | P = 0.349 |
| **Previous history of CS delivery** | | | | |
| Yes | 40(33.0) | 93(25.6) | 133(27.5) | $\chi2 = 2.519$ |
| No | 81(67.0) | 270(74.4) | 351(72.5) | P = .112 |
| **History of hypertension during last pregnancy** | | | | |
| Yes | 29(24.0) | 71(19.6) | 100(20.7) | $\chi2 = 1.076$ |
| No | 92(76.0) | 292(80.4) | 384(79.3) | P = 0.300 |
| Diagnosed with DM during last pregnancy | | | | |
| Yes | 12(9.9) | 41(11.3) | 53(11.0) | $\chi2 = 0.117$ |
| No | 109(90.1) | 322(88.7) | 431(89.0) | P = 0.674 |

cases and 46 (12.7%) from controls. The majority of women in cases (75%) and about half of the women in controls (186%) were non-autonomous in their decision-making (Table 4). Only 254 (52.5%) of respondents had a good practice of BPCR when it came to birth preparedness and complication readiness (BPCR). By regards to the percentages of specific BPCR components, 76.0% of cases and 72.4% of controls identified their place of birth, but only 13.2% of cases and 14.0% of controls identified blood donors if needed (**Fig 2**).

**Table 4. Maternal health service-related characteristics of mothers of neonates admitted to Public Hospitals in Hadiya Zone, Southern Ethiopia, 2021.**

| Variable categories | Cases = 121 | Controls = 363 | Total = 484 | Test statistics |
|---|---|---|---|---|
| | [n(%)] | [n(%)] | [n(%)] | |
| **ANC visit** | | | | |
| > = 4 | 30(24.8) | 150(41.3) | 180(37.2) | χ2 = 25.717 |
| 2–3 | 29(24.0) | 114(31.4) | 143(29.5) | P<0.001 |
| 1 | 46(38.0) | 81(22.3) | 127(26.2) | |
| No | 16(13.2) | 18(5.0) | 34(7.1) | |
| **Mode of delivery** | | | | |
| SVD | 71(58.7) | 295(81.3) | 366(75.6) | χ2 = 25.454 |
| Instrumental delivery | 14(11.6) | 22(6.1) | 36(7.4) | P<0.001 |
| C/S | 36(29.7) | 46(12.7) | 82(16.9) | |
| **Knowledge of danger signs** | | | | |
| Yes | 78(64.5) | 264(72.7) | 342(70.7) | χ2 = 2.990 |
| No | 43(35.5) | 99(27.3) | 142(29.3) | P = 0.084 |
| **Means of transportation** | | | | |
| On foot | 52(43.0) | 167(46.0) | 219(45.2) | χ2 = 0.455 |
| Rented transport | 41(33.9) | 121(33.3) | 162(33.5) | P = 0.797 |
| Ambulance | 28(23.1) | 75(20.7) | 103(21.3) | |
| **Autonomy in decision making** | | | | |
| Yes | 46(38.0) | 177(48.8) | 223(46.1) | χ2 = 4.21 |
| No | 75(62.0) | 186(51.2) | 261(53.9) | P<0.040 |
| **First Delay** | | | | |
| Yes (>24hr) | 73(60.3) | 180(49.6) | 253(52.3) | χ2 = 4.199 |
| No (≤24hr) | 48(39.7) | 183(50.4) | 231(47.7) | P = 0.040 |
| **Second delay** | | | | |
| Yes (>60min) | 48(39.7) | 101(27.8) | 149(30.8) | χ2 = 5.976 |
| No (≤60min) | 73(60.3) | 262(72.2) | 335(69.2) | P = 0.014 |
| **Third delay** | | | | |
| Yes(>60 min) | 72(59.5) | 108(29.8) | 180(37.2) | χ2 = 34.389 |
| No(≤60min) | 49(40.5) | 255(70.2) | 304(62.8) | P<0.001 |
| **Level of BPCR plan** | | | | |
| Good | 39(32.2) | 215(59.2) | 254(52.5) | χ2 = 26.523 |
| Poor | 82(67.8) | 148(40.8) | 230(47.5) | P<0.001 |

## Respondent's knowledge on neonatal danger signs

The nine WHO-UNICEF lists of newborn danger signs have been used to assess mothers' knowledge of these signs, and more than 7 out of ten respondents, 342(70.7%) had good knowledge of newborn danger signs, and the majority, 264(72.7%) were accounted by mothers of control groups. Unable to Breastfeed, 551(67.9%), and raised temperature, 518(63.8%), were the commonest danger sign mentioned by respondents (Fig 3).

## Clinical characteristics of neonatal near misses

The Latin American Centre for Perinatology (CLAP) definition of a neonatal near-miss was used to select cases. By the near-miss criteria, the pragmatic criteria took the lion's share of the two key criteria. Of the pragmatic criteria, the most prevalent newborn problem was gestational age less than 33 weeks, which accounted for 54 (44.6%), followed by birth weight less than 1750gm, 42 (34.7%). Of the management criteria, the use of intravenous antibiotics up to

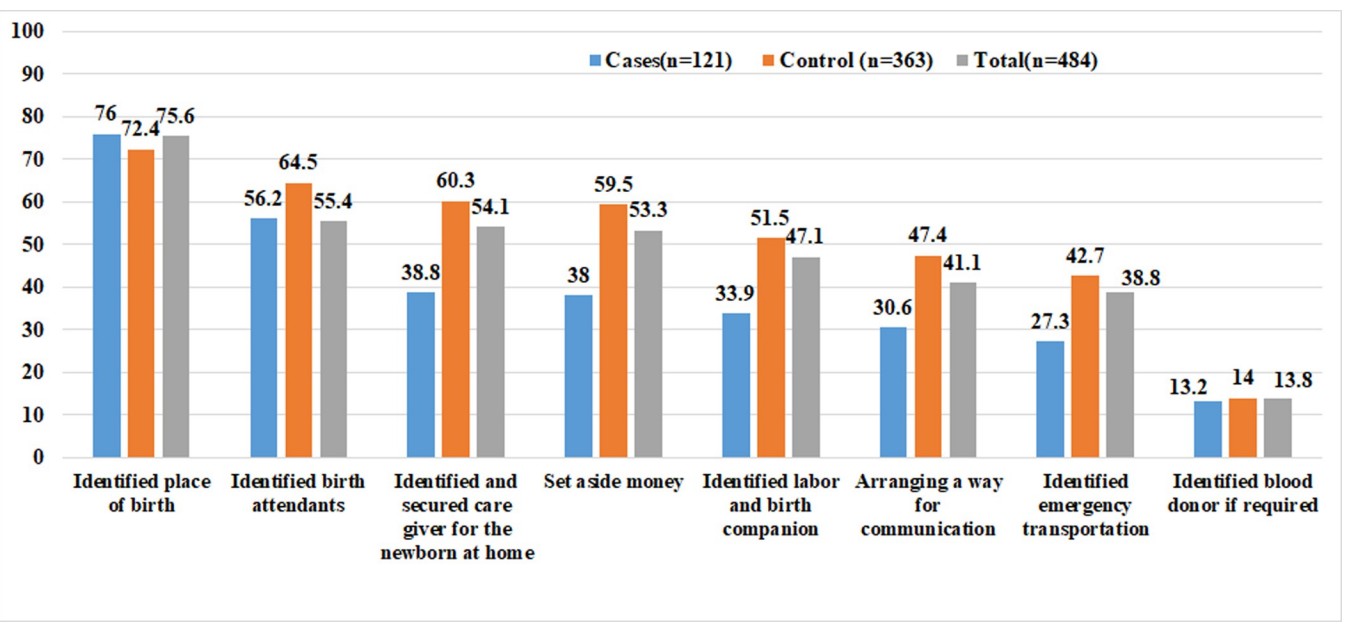

**Fig 2. Shows the percentages of BPCR practice of respondents in selected public hospitals of Hadiya Zone, Southern Ethiopia, 2021.**

7 days and before 28 days of life was experienced in the majority of cases 33(27.3%). There were no cases that experienced any surgical procedures and the use of corticosteroids for the treatment of refractory hypoglycemia (Table 5).

## Determinants of Neonatal near-miss (NNM)

In a multivariate logistic regression analysis, five variables were identified as significant determinants of NNM: birth interval of fewer than 24 months, lack of ANC, Cesarean mode of

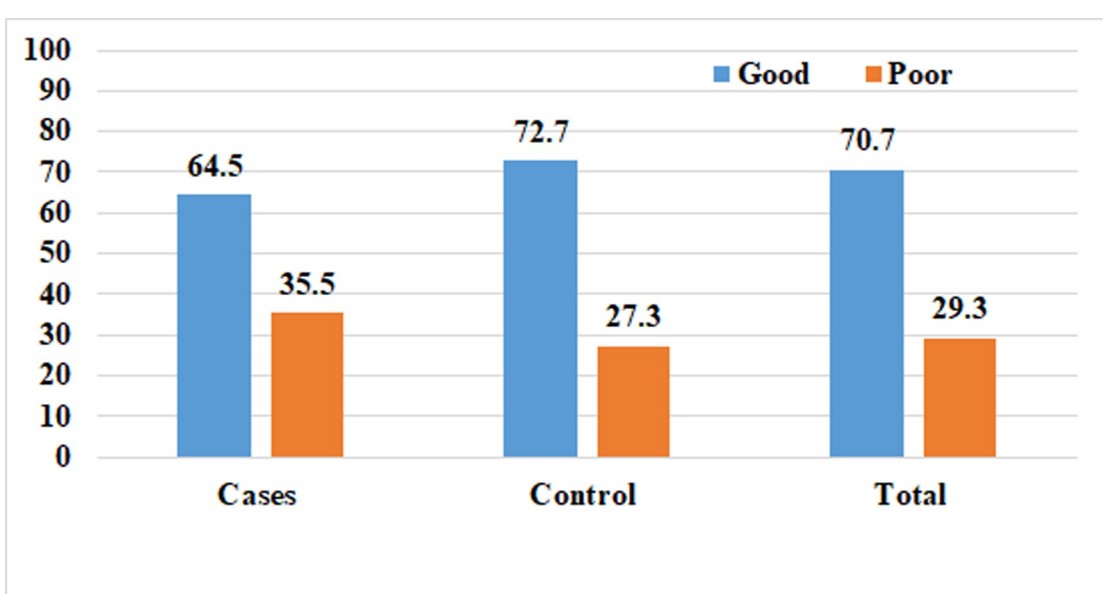

**Fig 3. Level of knowledge on Neonatal danger signs among mothers of neonates admitted to public hospitals in Hadiya zone, Southern Ethiopia, 2021.**

**Table 5. Clinical characteristics of neonatal near misses among neonates admitted to public hospitals of Hadiya zone, Southern Ethiopia, 2020.**

| Neonatal near-miss events(n = 121) | Frequency (%) |
|---|---|
| **Pragmatic criteria** | **97(80.1)** |
| APGAR score of less than 7 | 36(29.8) |
| Birth weight less than 1750g | 42(34.7) |
| Gestational age less than 33 weeks | 54(44.6) |
| **Management criteria** | **56(46.2)** |
| Cardiopulmonary resuscitation | 9(7.4) |
| Use of anticonvulsant | 4(2.3) |
| Use of phototherapy in the first 24 hours | 11(9.1) |
| Use of intravenous antibiotics up to 7 days and before 28 days of life | 33(27.3) |
| Use of corticosteroid for the treatment of refractory hypoglycemia | 0 |
| Nasal continuous positive airway pressure (NCPAP) | 13(10.7) |
| Any surgical procedure | 0(0.0) |
| Congenital malformation | 3(2.5) |
| Transfusion of blood derivatives | 4(2.3) |
| Any intubation | 13(10.7) |

delivery, sustaining a third maternal delay, and poor practice of birth preparedness and complication readiness (BPCR) plan.

NNM was found to be significantly affected by ANC follow-up. Women who did not have ANC follow-up had a 3.37 times higher risk of NNM than women who had four or more antenatal visits [AOR = 3.37; 95%CI: 1.35,6.39]. When compared to those who delivered via the normal vaginal route (SVD), neonates who delivered via cesarean section had a 2.24 times higher likelihood of being NNM cases [AOR = 2.24; 95%CI: 1.20,4.16]. The chance of being an NNM case is 2.15 times higher in neonates born with a short birth interval of fewer than 24 months compared to their counterparts [AOR = 2.15, 95% CI: 1.29,3.57].

Neonates born to mothers with a poor birth preparedness and complication readiness (BPCR) plan had a 2.5 times higher risk of NNM than those born to mothers with a good BPCR plan [AOR = 2.50; 95% CI: 1.49,4.13]. Furthermore, the risk of NNM was 3.47 times greater among mothers who experienced the third delay during their last birth compared to those who did not [AOR = 3.47; 95% CI: 2.11, 5.75] [Table 6].

## Discussion

Assessing cases of neonatal near-misses and identifying contributing factors can help to avoid neonatal death thoroughly and thoughtfully [23,28,39]. As a result, the goal of this study was to determine the factors that influence neonatal NNM in neonates admitted to public hospitals in southern Ethiopia. The lack of ANC, cesarean mode of delivery, the occurrence of a third maternal delay, and poor implementation of the birth preparedness and complication readiness (BPCR) plan were all identified as significant determinants of NNM in the current study.

The current study discovered that neonates with a birth interval of fewer than 24 months had a greater risk of having NNM than those with a birth interval of 24 months or more. Previously conducted studies from low- and middle-income countries identified a connection between newborn death and birth intervals of fewer than 24 months [46–48]. The birth interval effect in newborns could be linked to maternal nutritional depletion, which is caused by the mother's physiological competition with the growing fetus [49]. On the other edge, those with a shorter interval between conceptions are more likely to have an unwanted and

**Table 6. Determinants of NNM among mothers of neonates admitted in public hospitals in southern Ethiopia, Southern Ethiopia, 2020.**

| Variable | Neonatal near miss | | COR(95%CI) | AOR(95%CI) |
|---|---|---|---|---|
| | Cases (%) | Controls (%) | | |
| **Age of mothers in the year** | | | | |
| ≥35 | 19(15.8) | 59(16.3) | 1.74(0.59,5.15) | |
| 20–34 | 97(80.1) | 277(76.3) | 1.89(0.71,5.05) | |
| <20 | 5(4.1) | 27(7.4) | 1 | |
| **Residence** | | | | |
| Rural | 67(55.4) | 131(36.1) | 2.19(1.45,3.34)* | 1.54(0.93,2.53) |
| Urban | 54(44.6) | 232(63.9) | 1 | 1 |
| **Maternal Education** | | | | |
| No formal education | 46(38.0) | 116(31.9) | 2.07(1.06,4.02)* | 1.62(0.75,3.48) |
| Primary education | 31(25.6) | 87(24.0) | 1.86(0.92,3.75)* | 1.18(0.53,2.63) |
| Secondary education | 30(24.8) | 87(24.0) | 1.79(0.89,3.64)* | 1.59(0.71,3.55) |
| College and above | 14(11.6) | 73(20.1) | 1 | 1 |
| **Wealth index** | | | | |
| Lowest | 27(22.3) | 62(17.1) | 1.36(0.68,2.73) | 1.74(0.78,3.89) |
| Second | 29(24.0) | 74(20.4) | 1.96(1.02,3.75)* | 1.84(0.84,4.02) |
| Middle | 27(22.3) | 72(19.8) | 1.64(0.84,3.19)* | 1.78(0.82,3.83) |
| Fourth | 19(15.7) | 76(20.9) | 1.04(0.51,2.11) | 0.64(0.28,1.50) |
| Highest | 19(15.7) | 79(21.8) | 1 | 1 |
| **Family size** | | | | |
| ≥5 | 65(53.7) | 180(49.6) | 1.18(0.78,1.78) | |
| <5 | 56(46.3) | 183(50.4) | 1 | |
| **Sex of the newborn** | | | | |
| Male | 62(51.2) | 177(48.8) | 1.10(0.73,1.67) | |
| Female | 59(48.8) | 186(51.2) | 1 | |
| **Presentation during birth** | | | | |
| Non-vertex | 24(19.8) | 49(13.5) | 1.59(0.92,2.72)* | 1.89(0.98,3.64) |
| Vertex | 97(80.2) | 314(86.5) | 1 | 1 |
| **Parity** | | | | |
| 1(Primipara) | 17(14.1) | 66(18.2) | 1.34(0.68,2.62) | 1.44(0.66,3.15) |
| 2-4(Multipara) | 73(60.3) | 207(57.0) | 1.37(0.75,2.48) | 1.47(0.74,2.91) |
| ≥5(Grand multipara) | 31(25.6) | 90(24.8) | 1 | 1 |
| **Birth interval** | | | | |
| <24 months | 67(55.4) | 114(31.4) | 2.71(1.78,4.13)* | **2.15(1.29,3.57)**** |
| ≥24 months | 54(44.6) | 249(68.6) | 1 | 1 |
| **History of Neonatal death** | | | | |
| Yes | 18(14.9) | 26(7.2) | 2.26(1.19,4.29)* | 1.46(0.66,3.22) |
| No | 103(85.1) | 337(92.8) | 1 | 1 |
| **Previous history of CS delivery** | | | | |
| Yes | 40(33.0) | 93(25.6) | 1.43(0.92,2.24)* | 1.50(0.88,2.54) |
| No | 81(67.0) | 270(74.4) | 1 | 1 |
| **ANC visit** | | | | |
| No | 16(13.2) | 18(5.0) | 4.44(2.04,7.69)* | **3.37(1.35,6.39)**** |
| 1 | 46(38.0) | 81(22.3) | 2.84(1.67,4.84)* | 1.84(0.98,3.46) |
| 2–3 | 29(24.0) | 114(31.4) | 1.27(0.72,2.24) | 0.95(0.49,1.81) |
| ≥4 | 30(24.8) | 150(41.3) | 1 | 1 |
| **Mode of delivery** | | | | |
| C/S | 36(29.7) | 46(12.7) | 3.25(1.96,5.40) | **2.24(1.20,4.16)**** |

(*Continued*)

**Table 6.** (Continued)

| Variable | Neonatal near miss | | COR(95%CI) | AOR(95%CI) |
|---|---|---|---|---|
| | Cases (%) | Controls (%) | | |
| Instrumental delivery | 14(11.6) | 22(6.1) | 2.64(1.29,5.42) | 1.65(0.68,4.01) |
| SVD | 71(58.7) | 295(81.3) | 1 | 1 |
| **Knowledge of danger signs** | | | | |
| No | 43(35.5) | 99(27.3) | 1.47(0.95,2.28)* | 1.11(0.65,1.89) |
| Yes | 78(64.5) | 264(72.7) | 1 | 1 |
| **Having hypertension during the last pregnancy** | | | | |
| Yes | 29(24.0) | 71(19.6) | 1.07(0.65,1.768) | |
| No | 92(76.0) | 292(80.4) | 1 | |
| **Autonomy in decision making** | | | | |
| No | 75(62.0) | 186(51.2) | 1.55(1.02,2.36)* | 1.65(0.99,2.74) |
| Yes | 46(38.0) | 177(48.8) | 1 | 1 |
| **BPCR plan** | | | | |
| Poor | 82(67.8) | 148(40.8) | 3.05(1.98,4.72)* | **2.50(1.49,4.13)**** |
| Good | 39(32.2) | 215(59.2) | 1 | 1 |
| **Third delay** | | | | |
| Yes(>60 min) | 72(59.5) | 108(29.8) | 3.47(2.26,5.32)* | **3.47(2.11,5.75)**** |
| No(≤60min) | 49(40.5) | 255(70.2) | 1 | |

**Key:** 1: Reference category; AOR = Adjusted odds ratio, COR = Crude odds ratio

*Statistically significant at p-value<0.25

** Statistically significant at p-value <0.05.

unplanned pregnancy, and these women may not pay enough attention to their pregnancy or receive essential information such as dietary counseling and fetal monitoring. This exposes the fetus in the uterus to a variety of problems that later develop into severe neonatal morbidities (near-miss) [33]. These findings suggest that encouraging postpartum family planning could lower the number of newborn problems and deaths. Furthermore, because a mother's inter-birth interval was shorter, she didn't have enough time to prepare herself in terms of financial and material resources, which could result in a delay in service accessibility, ending in near-miss cases.

Furthermore, NNM was found to be significantly influenced by the frequency of ANC follow-up. This finding was supported by studies conducted in Eastern Brazil, Southern Ethiopia, and Southwest Ethiopia, which show that no prenatal care visits were the leading determinants of Neonatal Near Miss [32–34,40,50]. According to studies, having no or inadequate ANC visits during pregnancy has been linked to poor pregnancy outcomes due to a reduction in the provision and accessibility of health promotion on danger signs and postpartum complications [51,52]. This could be explained by the fact that no or insufficient ANC visits result in insufficient prenatal care, which alters the maternal continuum of care and, as a result, affects neonatal health outcomes [53]. On the other hand, not having antenatal care may limit women's access to information about possible danger signs during pregnancy and childbirth, which may fail to recognize deadly newborn conditions early and, as a consequence, NNM cases. As a result, it is highly suggested that adequate ANC should be provided as an essential input for reducing NNM cases, which is critical in minimizing neonatal death in the study area. Studies conducted in Brazil, Morocco, and southern Ethiopia, on the other hand, found no association between NNM and ANC follow-up [24,29,48,54].

Mode of delivery via cesarean section showed a significant association with NNM. This was in line with studies conducted in Brazil [13,24,32], South Africa [55], and Ethiopia [34,56].

Cesarean section delivery has been linked to increased newborn morbidity and mortality, as well as delayed or no improvement in neonatal outcomes [57]. Furthermore, cesarean section delivered newborns had less skin-to-skin contact with their mothers during the immediate postpartum period, which is critical for the newborn, and this could be accompanied by difficulties for neonates to breastfeed within one hour of birth, putting the neonate at a higher risk of early complications [58]. Likewise, a cesarean section on demand sometimes could be a risk factor for prematurity, which is one of the components of programmatic criteria [16]. These results suggested that health care providers should assess the potential risk of cesarean section and only perform it if there are compelling clinical justifications. To look at it another way, nonmedical grounds for cesarean section should be reduced to the WHO-recommended acceptable level (5–15%) to reduce neonatal health risks associated with cesarean section [59].

Neonates born to mothers with poor birth preparedness and complication readiness (BPCR) plans were more likely to be near-miss cases than those born to mothers with a good BPCR plan. This could be because women with a poor BPCR plan were more likely to experience maternal delays (such as delays in seeking, reaching, and receiving treatment) and all of the hastened NNM events [43]. This is a new finding in this study, and it has policy implications because BPCR is one of the WHO's twelve major recommendations for increasing the use of skilled maternity care and reducing dangerous obstetric problems by using facility care at the right time [44]. Complication readiness also engages the woman, her family, the community, and health care providers in proactive health services by equipping them to spot early danger signs of pregnancy and childbirth, as well as provide emergency obstetric care (EOC). As a result, a concerted effort from health care providers at the community (HEWs) and facility levels is required to improve BPCR practice from conception to delivery.

Third delay(Delay in obtaining adequate and appropriate treatment while the mother arrived in a health facility) was a significant determinant of NNM. This finding was backed up by a study conducted in Brazil, which indicated that the third delay contributed significantly to maternal and newborn risks [60,61]. Lack of qualified and skilled personnel, insufficient staff, limited availability of medicine and equipment, generally poor conditions of the facilities, and poor attitudes and treatment on the part of medical workers are all possible reasons for the delay, and stakeholders working on maternal and neonatal health should place a strong emphasis on overcoming these impediments [45].

The most important aspect of this study for public health is that it identifies potential characteristics that predispose newborns to life-threatening (near-miss) conditions, which is critical to address the underlying causes and provide prompt remedies by various stakeholders in the healthcare system. This study will be useful to health policymakers and program developers when it comes to newborn health in the healthcare system. Also, the study used validated and standardized Neonatal Near Miss identification criteria to avoid misclassification and unlike most of the recently conducted studies, it tried to assess the effect of the three delays on NNM. Despite its strengths, this study contains the following limitations. Although the reported cases were verified by senior experts, there may be a misclassification bias. Confounders are difficult to control since cases and controls are not matched with relevant variables due to the study design. The respondents may be prone to social desirability bias because the study was based on self-reports. Finally, there is a possibility of recall bias because women were asked about occurrences that occurred within the previous year before this study.

## Conclusion

The current study identified a lack of ANC, cesarean delivery, the occurrence of a third maternal delay, and poor implementation of the birth preparedness and complication readiness

(BPCR) plan as significant determinants of NNM. The provision of adequate ANC should be a priority for health care providers at static and outreach service delivery points. To avoid serious neonatal problems, mothers who deliver by Cesarean section should receive more attention from their families and health care providers. Health care providers in the ANC unit should encourage pregnant women to implement the WHO-recommended elements of the BPCR plan. To achieve optimal birth spacing, healthcare managers and providers should focus on contraceptive provision. Unnecessary delays in health facilities during childbirth should be avoided at all costs.

## Supporting information

**S1 Data. Data collection tool used to identify determinants of neonatal Near miss.**
(DOCX)

**S1 Dataset. The raw data supporting the findings of this study.**
(SAV)

## Acknowledgments

We are grateful to Wachemo University College of Medicine and Health Science, School of Public Health for providing Ethical approval for this research. We are indebted to the managers and healthcare professionals who worked in the selected hospitals for their assistance and support during the study. Finally, for their efforts, we want to thank our supervisors, data collectors, and study participants.

## Author Contributions

**Conceptualization:** Aklilu Habte.

**Data curation:** Addisalem Gizachew.

**Formal analysis:** Aklilu Habte.

**Investigation:** Kaleegziabher Lukas, Tamirat Melis, Mulugeta Hailu, Addisalem Gizachew.

**Methodology:** Aklilu Habte, Kaleegziabher Lukas, Aiggan Tamene, Addisalem Gizachew.

**Project administration:** Aklilu Habte.

**Software:** Aklilu Habte.

**Supervision:** Aklilu Habte, Kaleegziabher Lukas, Tamirat Melis, Aiggan Tamene, Tadesse Sahle, Mulugeta Hailu.

**Validation:** Aklilu Habte.

**Visualization:** Aklilu Habte.

**Writing – original draft:** Aklilu Habte.

**Writing – review & editing:** Aklilu Habte, Kaleegziabher Lukas, Tamirat Melis, Aiggan Tamene, Tadesse Sahle, Addisalem Gizachew.

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
