## [Decision Letter · Decision Letter 0]

28 Mar 2022

PONE-D-21-29699Determinants of Neonatal Near Miss among Neonates Admitted To Public Hospitals of Southern Ethiopia, 2021: A Case-Control StudyPLOS ONE

Dear Dr. Habte,

Thank you for submitting your manuscript to PLOS ONE. After careful consideration, we feel that it has merit but does not fully meet PLOS ONE’s publication criteria as it currently stands. Therefore, we invite you to submit a revised version of the manuscript that addresses the points raised during the review process.

We look forward to receiving your revised manuscript.

Kind regards,

Devendra Raj Singh, MSc Health Promotion & Public Health, MA

Academic Editor

PLOS ONE

Reviewers' comments:

Reviewer's Responses to Questions

**Comments to the Author**

1. Is the manuscript technically sound, and do the data support the conclusions?

Reviewer #1: Yes

Reviewer #2: Yes

Reviewer #3: Yes

Reviewer #4: Yes

2. Has the statistical analysis been performed appropriately and rigorously? 

Reviewer #1: Yes

Reviewer #2: Yes

Reviewer #3: Yes

Reviewer #4: Yes

3. Have the authors made all data underlying the findings in their manuscript fully available?

Reviewer #1: Yes

Reviewer #2: Yes

Reviewer #3: Yes

Reviewer #4: Yes

4. Is the manuscript presented in an intelligible fashion and written in standard English?

Reviewer #1: Yes

Reviewer #2: Yes

Reviewer #3: No

Reviewer #4: No

5. Review Comments to the Author

Reviewer #1: 1. Sample size calculation…..

• Why did you take the percentage of cases exposed to old maternal age …….? What is the linkage between your variables

2. Ethical consideration and consent to participate….

• Do you think issuing unique ID numbers ensures confidentiality?

3. SOCIODEMOGRAPHIC…(TABLE 1)

• Age of the mother ….calculate x2

4. Characteristics of the newborns

• Line 4 ……….The majority of cases (80.2%) and controls (314.5%) had a vertex presentation during…. Needs correction.

5. Maternal health service-related characteristic

• How do you measure birth preparedness and complication readiness (BPCR)? To classify the variable as good and poor?

6. Table5. Determinants of NNM among Mothers of neonates admitted in public hospitals in southern Ethiopia, Southern Ethiopia, 2020.

• Why you did not analyzed AOR variables maternal age, family size, sex of the newborn and Having hypertension during the last pregnancy

Reviewer #2: This an interesting paper which clearly indicates the major public health importance Neonatal near miss in the low income country particularly Ethiopia. The paper is worth for publication after correcting the minor comments.

1- There are several grammatical problems throughout the document, which requires extensive English language editing. Professional English editing is needed

2- In the abstract part the authors used introduction, I believe better replace the word introduction by background

3- In the abstract part the authors describe conclusion and recommendation as Stakeholders at the zonal and regional levels need to step up their efforts to address the barriers that prevent health facilities from providing adequate and appropriate care. Furthermore, to prevent major neonatal problems, women who have not had an ANC and who deliver by Cesarean section require closer attention from their family and health care providers. This is general unclear for the reader. Better to make the a bit specific the conclusion and recommendation inline with your findings.

4. In the method part your sampling procedures is very short. would you describe the sampling procedure?

5. I was wondering that if you more explain The pragmatic and management criteria?

6- In the method section, data collection tools, could you more describe more about your measurement instrument validity.

7- In the method part Sampling procedures for the selection of cases you have used consecutive sampling and for the selection of controls systematic sampling is used. what do you think about the generalization?

8- in the data analysis technique you have deal with The Hosmer and Lemeshow goodness of fit test was used to assess the model's fitness. The variance inflation factor (VIF) was used to check for multicollinearity among independent variables. You should have to put the value obtained from the data analysis output with the respective interpretation.

Reviewer #3: Dear authors thank you very much for writing this very interesting topic. Neonatal mortality in Ethiopia is unacceptably high. These identifying determinant factors of neonatal near miss will be helpful for policy makers and stakeholders. Saying this I have listed my comments below.

Abstract

1. In the Introduction section you define Neonatal near-miss (NNM) between the ages of 0 and 27 days. What about 28 days, in which group of age we called neonates?

2. Rather than saying southern Ethiopia, it is better to write the specific area

3. Why you select unmatched case control study, why not matched case control study?

4. What is third maternal delay, be specific which delay?

5. How do you measure poor birth preparedness and complication readiness?

6. You finding and your conclusion is very far apart. For instance have studied barriers of health facilities? Are studied health care workers at community level, you have listed all in the recommendation. I think it needs rewritten.

Background

1. It needs synthesis, chronologic order

2. Several studies were conducted regarding neonatal near miss, however you were not addressed their limitation and the need of your study.

Sample size determination

1. Very critical why select a study conducted in Brazil to calculate your sample size; there were many of study in Ethiopia?

2. Who select the cases?

Discussion

Well written

Reviewer #4: The manuscript has been written in a detail form. As well, it’s great public health importance. However, correct major problems such as coherence and paragraph structuring, and improve grammar flow. Overall, incorporate the specific comments forwarded in the word document.

6. PLOS authors have the option to publish the peer review history of their article (what does this mean?). If published, this will include your full peer review and any attached files.

Reviewer #1: No

Reviewer #2: No

Reviewer #3: No

Reviewer #4: No

---

## [Author Response · Author response to Decision Letter 0]

1 Apr 2022

The response to each reviewer's comment have been attached as a "Response to Reviewers" in the submission process

---

## [Decision Letter · Decision Letter 1]

21 Apr 2022

Determinants of Neonatal Near Miss among Neonates Admitted to Public Hospitals of Southern Ethiopia, 2021: A Case-Control Study

PONE-D-21-29699R1

Dear Dr. Habte,

We’re pleased to inform you that your manuscript has been judged scientifically suitable for publication and will be formally accepted for publication once it meets all outstanding technical requirements.

Kind regards,

Devendra Raj Singh, MSc Health Promotion & Public Health, MA

Academic Editor

PLOS ONE

Additional Editor Comments (optional):

Reviewers' comments:

Reviewer's Responses to Questions

**Comments to the Author**

1. If the authors have adequately addressed your comments raised in a previous round of review and you feel that this manuscript is now acceptable for publication, you may indicate that here to bypass the “Comments to the Author” section, enter your conflict of interest statement in the “Confidential to Editor” section, and submit your "Accept" recommendation.

Reviewer #1: All comments have been addressed

Reviewer #3: All comments have been addressed

Reviewer #4: All comments have been addressed

2. Is the manuscript technically sound, and do the data support the conclusions?

Reviewer #1: Yes

Reviewer #3: Yes

Reviewer #4: Yes

3. Has the statistical analysis been performed appropriately and rigorously? 

Reviewer #1: Yes

Reviewer #3: Yes

Reviewer #4: Yes

4. Have the authors made all data underlying the findings in their manuscript fully available?

Reviewer #1: Yes

Reviewer #3: Yes

Reviewer #4: Yes

5. Is the manuscript presented in an intelligible fashion and written in standard English?

Reviewer #1: Yes

Reviewer #3: Yes

Reviewer #4: Yes

6. Review Comments to the Author

Reviewer #1: (No Response)

Reviewer #3: Dear Author, thank you for submitting the revised version of the manuscript. All of my comments have been addressed.

Reviewer #4: (No Response)

7. PLOS authors have the option to publish the peer review history of their article (what does this mean?). If published, this will include your full peer review and any attached files.

Reviewer #1: No

Reviewer #3: No

Reviewer #4: No

---

## [Editor Report · Acceptance letter]

29 Apr 2022

PONE-D-21-29699R1 

Determinants of Neonatal Near Miss among Neonates Admitted to Public Hospitals in Southern Ethiopia, 2021: A Case-Control Study 

Dear Dr. Habte:

I'm pleased to inform you that your manuscript has been deemed suitable for publication in PLOS ONE. Congratulations! Your manuscript is now with our production department. 

Kind regards, 

on behalf of

Mr. Devendra Raj Singh 

Academic Editor

PLOS ONE